# Multiple Groups of Agents for Increased Movement Interference and Synchronization

**DOI:** 10.3390/s22145465

**Published:** 2022-07-21

**Authors:** Alexis Meneses, Hamed Mahzoon, Yuichiro Yoshikawa, Hiroshi Ishiguro

**Affiliations:** 1Graduate School of Engineering Science, Osaka University, Toyonaka 560-8531, Japan; yoshikawa@irl.sys.es.osaka-u.ac.jp (Y.Y.); ishiguro@sys.es.osaka-u.ac.jp (H.I.); 2Institute for Open and Transdisciplinary Research Initiatives (OTRI), Osaka University, Suita 565-0871, Japan; hamed.mahzoon@irl.sys.es.osaka-u.ac.jp

**Keywords:** interference, synchronization, virtual agent, human–robot interaction, human–agent interaction

## Abstract

We examined the influence of groups of agents and the type of avatar on movement interference. In addition, we studied the synchronization of the subject with the agent. For that, we conducted experiments utilizing human subjects to examine the influence of one, two, or three agents, as well as human or robot avatars, and finally, the agent moving biologically or linearly. We found the main effect on movement interference was the number of agents; namely, three agents had significantly more influence on movement interference than one agent. These results suggest that the number of agents is more influential on movement interference than other avatar characteristics. For the synchronization, the main effect of the type of the agent was revealed, showing that the human agent kept more synchronization compared to the robotic agent. In this experiment, we introduced an additional paradigm on the interference which we called synchronization, discovering that a group of agents is able to influence this behavioral level as well.

## 1. Introduction

The development of novel communication technologies, e.g., humanoid robots [1,2,3] and virtual reality avatars [4,5,6], influence social behavior in society [7]. A type of influential social behavior was shown in human kinematic performance, which can be influenced by observing incongruent movement of different humans [8], naming this type of influence “interference”. Analyzing the potential factors that produce this interference is fundamental to understanding the effect of emerging technologies on humans. The interference effect has been studied under different conditions, including observing humans on screens, interacting with robots, and in virtual reality [9,10]. The present study explores the influence of a group of agents, such as the type of movement of the agent and the virtual agents’ avatars.

Virtual avatars have also been shown to influence humans’ shopping decisions and behaviors in the virtual world, depending on the avatar shape [11,12]. Moreover, previous studies reported that the shape of a virtual avatar changes the perception of someone’s weight [13] and influences the kinematics movement interference [14]. Not only the shape, but the size of the avatar may influence the perception of the avatar [15]. In addition, and importantly, the movement of the avatar may also contribute to a perception of the avatar as being more realistic [16]. Thus, finding appropriate embodiment avatars plays a crucial role in increasing motion interference.

Not only avatars may influence the kinematics movement interference. It is known that groups of people influence members of their groups in their behavior and perception of others [17]. For example, those in a group tend to assume riskier behaviors than when they are alone [18]. Groups of agents are also influential in terms of group perceptions regarding decisions such as punishing others even when they do not want to do it [19]. Moreover, conversational groups of two robots enable people to follow the conversation more easily than single robots [20]. Further, there is evidence that groups of robots can synchronize with humans in musical environments [21]. Synchronization is well known to enhance empathy and feelings of closeness to others [22,23]. Thus, the influence of a group of synchronized agents can be greater than the influence of a single agent.

Timing action synchronization is a feature that humans can naturally develop, e.g., humans naturally synchronize their claps in environments where there is a group of humans clapping [24]. Even though humans can naturally develop synchronization, it is not always easy to synchronize with others, e.g., for humans that have problems concentrating or cannot follow the beat in music [25,26]. Synchronization in different environments depends on the task the user is performing, e.g., the ability to synchronize depends on the muscles of humans [27]. Some actions, especially in a game, require coordination that does not require having the same starting timing movement [28]. This complementary synchronization depends on the phase of the agents’ time, meaning the more the phases are kept constant but not the same, the better performance they have [29]. To explore the influence of virtual agents on different types of synchronization, we propose to use antiphase synchronization movements, defining it as the ninety degree phase deviation.

The interference effect has not previously been studied using a group of synchronized agents or the type of avatar in this study. Adding a group of agents in virtual reality environments may influence the result of the movement interference in groups. Moreover, these agents may be able to enhance the synchronization timing of the movement of the hand. Exploring the influence of an avatar’s characteristics on a group of synchronized agents will help to design more influential systems of virtual agents in the future. In this study, two different projection groups of avatars were used to increase motion interference.

## 2. Related Work

Movement interference is a phenomenon that has previously been studied using humans or robots. However, these studies did not investigate the effects of the interference of groups (number of agents in groups) and the type of agents (human or robot). Movement interference appears in many degrees, as William James described that imaging performing a movement awakens the possible movement to some degree [30] In other studies, it was verified that imaging an action, observing someone’s action, or trying to represent the action in a person’s self-mind excites the muscles to be used to execute the action imagined, observed, or tried to be represented [31,32]. This phenomenon leads to the assumption that the action performance of a person may be able to be influenced by other humans’ performance. For example, a person’s hand muscle potential is linked to the observation of an active movement, while being the observer of a different person remains linked temporally to the observed movement [33]. In addition, the timing of the muscle initiation of a person becomes slower after watching another person moving their finger in similar patterns, or even when the other person is grasping an object in a similar way to the person [34,35]. Moreover, the actions of a person related to the time it takes to perform the action is another characteristic. Wild found that people tend to imitate models’ actions when they perform a synchronous action [36]. In this study, he asked the subjects to move their pointer finger on a table with points at a certain speed while watching a video of a human hand moving. He found that the imitation accuracy regarding time increased when the cognitive demand of the action was less intensive. In his study, he suggested that non-goal actions are more influential on visuomotor mapping (interference movement) in contrast to goal action which is more influential on accuracy. Hayes extended Wild’s work by creating a different task with mouse pointers [37]. Hayes drew two red points on a monitor screen and showed videos before the experiment started where the mouse cursor moved from one point to the other one. He manipulated the videos to show atypical movement speeds on the cursors, assuming that when the cursor moves more human-like, the cursors will be able to move similar to the human-like movements. He found that in cursor movements, the people were more willing to learn the imitation the movement when the movement was more human-like. None of these studies explore whether a group of agents may be able to influence the action of the agent. Moreover, the evaluation of a high cognitive demand task was not studied enough. Studying a task such as people trying to keep asynchronous movement may clarify the human movement interference.

Movement interference has been studied on avatars as well. Kupferberg found that movement interference depends on the similarity of the movement that the agents have [38]. In this study, she prepared a robotic arm that moves in front of a human with orthogonal and parallel movements. They used four stimuli: a human, a humanoid torso, a robotic hand with a table base, and a robotic hand with the base attached to the wall. They suggest that the configuration of the speed is more important than the presence of human-like features, such as body shape. In this study, they also suggest that the biological movement of the human-like hand may not influence the interference. In opposition, Chamide [39] suggested that humanoid robots (without faces or any social stimuli) may be more influential by having anthropomorphic body shapes. He found that human-like movement speed was significantly more influential than robot-like movement speed. A similar conclusion was proposed by Gandolfo [40], who compared a virtual humanoid agent with a human-like body shape, a real human, and a non-humanoid agent. The task was to grasp an object instead of moving a hand. They compared the position variance of the hand at grasping and found no significant difference in the results. None of these studies tried to compare the social aspects of the avatars, but focused on the movement, and they did not consider whether the conception of the avatar would influence the movement of the subject to be evaluated.

## 3. Materials and Methods

To investigate the effect of multiple agents on movement interference, we conducted an experiment in which we investigated combinations of the following variables: number of agents (one, two, and three), type of avatar (human and robot), and agent behavior (human-like agent moving with biological movement and robot-like agent moving with linear movement). We made the agents move their hands orthogonally to the movement of the subjects to influence the subjects’ movements. Each group of variables was separated depending on the subject movement, i.e., vertical hand movement or horizontal hand movement. In this experiment, we hypothesize that (h1) the higher the number of agents, the higher movement interference they influenced, (h2) the biological moving agent interfered more than the linear moving agent, and (h3) the human agent interfered with the movement more than the robot agent. We additionally hypothesize that (h4) the higher the number of agents, the higher the influence on the synchronization between the virtual agents and a person, (h5) the biological movement agent influences the synchronization between the projected virtual agents and a person more than the linear moving agent, and (h6) the human agent influences the synchronization between the projected virtual agents and a person more than the robot agent. The subjects provided written consent, and the experiments were approved by the ethical committee.

### 3.1. Subjects

A total of 24 subjects participated in all the conditions (13 females and 11 males, mean age = 22.17, standard deviation = 3.69). The order of conditions was counterbalanced for the number of agents, type of avatar, and agent behavior. After the orthogonal movement, the synchronization movement conditions were prepared.

### 3.2. System

We developed a system for tracking hand movement. We used two Optitrack V120 trio desktop trackers for recording the position of passive infrared markers with a frequency of 120 Hz. The Optitrack cameras were placed side by side with the distance of 3 m from each other and put on a cylinder bar with the height of 1.8 m. The trio camera had 3 camera sensors, with a shutter speed of 1 ms, a focal lens of 3.5 mm, and a low pass filter of 800 nm IR. The measurement error of the hand position of the participants was around 0.8% based on the frequency of the cameras.

We prepared two client programs to record every point after synchronizing their timing by using network time synchronization. As the sensors did not have an NTP server, we needed to estimate the package delay after some seconds of tracking the points to be able to recover the possible missing positions due to the loss of focus of the markers in front of the cameras. We tracked the first 10 packages of each Optitrack, calculated the time difference between them, and calculated the time average for having a time delay between the cameras. This could be explained with the following formula:(1)$$ɣ=\frac{\sum_{i=0}^{10}{\text{µ}}_{i}-\sum_{j=0}^{10}\zeta_{j}}{10}\;\;\;\;$$
where: ɣ : The average delay time;${µ}_{i}$: The delay time between tracked point on the Optitrack1;$\zeta_{i}$: The delay time between tracked point on the Optitrack2.

The average delay time was used for calculating the difference between the receiving time and the estimated real-time measurement. This estimated real-time measurement was stored on a variable in the program before the experiment started. The starting recording of the tracking point was received by an additional control socket which was developed by using socket.io on a Nodejs server (Figure 1).

The robot model was displayed by using the three.js library. The library allows moving the robot remotely by using WebSockets. We created an additional signaling server to receive the command from the experimenter to start the condition required for the experiment. After the experiment finished, there was a function created to send a message to the recording program to store the data on a JSON file by using Nodejs information. The algorithm of the system is shown in Figure 2.

As the robot needed to have biological movement, we recorded a human model’s movement to simulate the biological movement in the robot. The robots were able to move their hands in synchronization due to the time attribute javascript scope shared between the robot model displayed on the screen system.

### 3.3. Experimental Set-Up

Two Optitrack cameras were placed one in front of the other on cylinder bars facing the passive retroreflective marker. We placed a chair 1.2 m from the wall. Before the experiment, a background image was projected on the wall. The projected image had a resolution of 1280 × 720 pixels and a size of 2 m in width and 1.125 m in height. The environment set-up is shown in Figure 3. We recorded the movement of the marker using a Python script that connected through TCP protocol to the Optitrack software and stored the movement of the tracker on two separate files.

### 3.4. Avatar of the Agents

We recorded one colleague for projecting a human avatar on the wall. The size of the human avatar was 80% of the height of the projection window on the wall (0.9 m) and 0.3 m in width. Figure 4 shows an example of the projection of the human avatar. The human avatar was created using a video of a human and edited frame by frame to create a constant speed, naming it linear movement.

We used a robotic model to project the robot avatar on the wall. The size of the robot was similar to the human avatar. The robot was a humanoid robot based on a robot named “CommU”, developed by Vstonre Co., Ltd., Osaka, Japan, in collaboration with Osaka University. The projected robot head employed 3 degrees of freedom (DoF), two eyes with 3 DoF, upper eyelids with 1 DoF, a mouth with 1 DoF, two arms, each with 2 DoF, and a waist with 2 DoF. This model was manipulated using Three.js and WebSockets. Figure 5 shows an example of the projection of the robot avatar. The movement of the robot was tracked to a human avatar for creating the biological movements of the robot.

The controlling of the avatar movements was adjusted to start at the same time and end at the same time. The human avatar was able to move at the same speed as the robotic avatar for biological and linear movement.

### 3.5. Procedure

All instructions, including informed consent, were provided via written documentation. The subjects’ tasks were explained before the experiment. The subjects were instructed to move their hands vertically and horizontally on hearing a metronome at 2 Hz (120 bpm) before the experiment started. The training session consisted of 20 sessions (5 s for each session) with a total training time of 5 min. Before each set-up in the experiment, we instructed the subject to move their hands in vertical or horizontal directions. We played an intermittent beep sound at 2 Hz for 3 s at the beginning of each session, muting it after the 3 s. We instructed the subjects to start their hand movements after hearing two beeps, asking them to synchronize their movement with the third beep. The projection images were displayed after the sixth beep. At the end of the experiment, a beep sound was played to make the subjects stop. Subjects moved their hands for 15 s in each set-up, in a total of 24 conditions: 2 types of avatars (robot/human), 2 types of movement (biological movement, linear movement), number of agents (one/two/three), and 2 movement directions (vertical/horizontal). Additionally, the subjects attended to 24 other conditions on movements corresponding to the parallel movement of the hand. The subjects started at opposite movement times by adding an extra beep to the 3 s initial condition beeps and having the same independent variables of the kinematics movements. The movement of the hand started on a 90 degree difference in phase; due to this, we named this antiphase movement. Following each set-up, subjects were allowed to rest for 15 s. If the subject felt fatigued, the experimenter stopped the experiment for 5 min to allow the subject to recover from fatigue. We asked one random question regarding physical feeling in the subjects’ hands at the end of every set-up. Figure 6 shows an example of the movement recorded by the subject.

As Figure 7 shows, we combined the signals obtained from the cameras and calculated the standard deviation of the orthogonal axis.

The data obtained on the position of the hand were processed after the experiment finished. We combined the signals for compensating the possible signal loss. We evaluated the orthogonal coordinate of the movement of the agent for the interference movement and the parallel movement of the agent for the antiphase synchronization.

### 3.6. Evaluation on Interference Movement

We evaluated and compared the kinematics interference movement of the subject’s hand in each condition. As we utilized two cameras to avoid loss, we combined the signals and processed them to have the first 14 movements of the hand, as we assumed that fatigue developed after 7 s of moving the hand due to the occlusion of the markers. We processed the incongruent axis as Formula (2) shows.
(2)$$\sigma_{T}=\sqrt{\frac{\sum \left(x_{i}-\mu \right)}{N}}\;\;\;\;$$
where:$\sigma_{T}$: Standard deviation of the position of the incongruent axis;$x_{i}$: Position of the incongruent axis on time *i*;μ: Mean of the position of the incongruent axis;N: Number of sample analyzed.

In other words, the interference movement was the standard deviation of the incongruent axis of the position of the marker of the hand.

### 3.7. Evaluation on Synchronization

Evaluating the antiphase synchronization in time requires a measurement based on the phase changes. As we were interested in understanding the changes in the synchronization, we projected the cycle starting time and calculated the cycle phase value of an expected agent’s trajectory. By calculating the cycle phase changes, we were able to evaluate the variation in the synchronization between agent and subject.

We evaluated the changes in the synchronization based on the peaks of the subject trajectory hand position. As we utilized two cameras to avoid loss, we followed by combining the signals of both cameras. We processed them to have the first 14 movements of the hand, as we assumed fatigue was developed after 7 s of moving the hand due to the occlusion of the passive market we placed on the subject.

We created a sinusoidal wave that started on the antiphase movement of the subject’s first movement (ninety degree phase) and named it as an expected trajectory of the agent. To calculate the synchronization between the subject and the agent, we projected the peak values of the trajectory subject’s hand position into the expected trajectory agent’s hand position so that the first peak projection was in the minus ninety degree phase of the trajectory agent’s hand position. In order to obtain the phase value for each correspondent agent peak on the periodic movement of the subject’s hand, the inverse sinusoidal function of the subject indicated that time corresponding to the agent peak was calculated as Formula (3) shows. Figure 8 shows an example of a standardized subject’s hand trajectory and the projection of the peaks on the y plane or x plane for the vertical or horizontal movement, respectively.
(3)$$\theta_{s}=\sin^{-1}\left(P\left(t\right)\right)\;\;$$
where:$\theta_{s}$: Phase of subject in agent movement peaks;$P\left(t\right)$: Position of the subject’s hand at time *t*;t: Agent phase position at ninety degrees.


The signals were divided into two sections by separating the phase projections of the first half time and the latter half time, as Formula (4) shows.
(4)$$Si=\frac{\sum_{i=0}^{n}\theta_{i}}{n}-\frac{\sum_{j=0}^{m}\gamma_{j}}{m}\;\;\;$$
where:$\theta_{i}$: Phase on the prior half time *i*;n: Number of phases corresponding to peaks in the prior half;$\gamma_{j}$: Phase on the latter half time;m: Number of phases corresponding to peaks in the latter half;Si: Synchronization index.

As Figure 9 shows, we combined the signals obtained from the cameras and calculated the synchronization index of the parallel axis.

The difference between the average of the phases in the first half and the latter half was the synchronization index.

## 4. Results

In the following result report, M means mean, SD means standard deviation, and *p* is the calculated probability of alpha of the utilized *t*-test.

### 4.1. Results on Movement Interference

We conducted a three-way ANOVA for the horizontal movement interference, the results of which are shown in Table 1. We did not find a significant interaction effect on the variables. We found the main effect (F (2, 276) = 3.72, *p* < 0.05) in the number of robots. Then, post hoc analysis utilizing Bonferroni correction (here, adjusted alpha levels of 0.017) was conducted (Figure 10). The results show that the three-agent set-up has greater interference on the movement (M = 33.38 mm, SD = 19.15 mm) than the one-agent set-up (M = 27.3 mm, SD = 14.86 mm), where t (190) = 2.46, *p* = 0.015, and Cohen’s d = 0.35. In contrast, the two-agent set-up (M = 28.46 mm, SD = 13.98 mm) was not significantly different from the one-agent set-up (M = 27.3 mm, SD = 14.86 mm), where t (190) = 0.56, *p* = 0.578, and Cohen’s d = 0.08; similarly, the three agent set-up (M = 33.38 mm, SD = 19.15 mm) was not significantly different from the two-agent set-up (M = 28.46 mm, SD = 13.98 mm), where t (190) = 2.03, *p* = 0.043, and Cohen’s d = 0.29.

We conducted a three-way ANOVA for vertical movement in movement interference, of which the results are shown in Table 2. We found the main effect (F (2, 276) = 4.49, *p* < 0.05) in the number of robots. Then, post hoc analysis utilizing Bonferroni correction (here, adjusted alpha levels of 0.017) was conducted (Figure 11). The results show that the three-agent set-up has greater interference on the movement (M = 18.82 mm, SD = 9.28 mm) than the one agent set-up (M = 15.49 mm, SD = 6.76 mm), where t (190) = 2.85, *p* = 0.005, and d = 0.41. In contrast, the two-agent set-up (M = 16.72 mm, SD = 6.93 mm) was not significantly different from the one-agent set-up (M = 15.49 mm, SD = 6.76 mm), where t (190) = 0.18, *p* = 0.215, and Cohen’s d = 0.18; similarly, the three-agent set-up (M = 18.82 mm, SD = 9.28 mm) was not significantly different from the two-agent set-up (M = 16.72 mm, SD = 6.93 mm), where t (190) = 1.78, *p* = 0.076, and Cohen’s d = 0.26.

### 4.2. Results on Time Synchronization

We conducted a three-way ANOVA for the vertical time synchronization, the results of which are shown in Table 3. We did not find a significant interaction effect on the variables. We found the that main effect (F (1, 276) = 10.63, *p* < 0.01) was on the agents’ avatar. Then, post hoc analysis utilizing Bonferroni correction (here, adjusted alpha levels of 0.05) was conducted (Figure 12). The results show that robot has a greater phase change in the hand movement (M = 44.66 deg, SD = 33.24 deg) than the human agent set-up (M = 31.68 deg, SD = 33.64 deg), where t (286) = 3.29, *p* < 0.01, and Cohen’s d = 0.39. In contrast, there was no significant main effect for the agent number (F (2, 276) = 0.84, *p* = 0.43); similarly, there was no significant main effect for the biological movement (F (1, 276) = 1.06, *p* = 0.3).

We conducted a three-way ANOVA for the horizontal time synchronization, the results of which are shown in Table 4. We did not find a significant interaction effect on the variables. We found that the main effect (F (1, 276) = 10.63, *p* < 0.01) was on the type of avatar. Then, post hoc analysis utilizing Bonferroni correction (here, adjusted alpha levels of 0.05) was conducted (Figure 13). The results show that the robot has a greater phase change in the hand movement (M = 34.72 deg, SD = 30.99 deg) than the human agent set-up (M = 22.82 deg, SD = 23.18 deg), where t (286) = 3.69, *p* < 0.01, and Cohen’s d = 0.39. In contrast, there was no significant main effect for the agent number (F (2, 276) = 0.12, *p* = 0.89); similarly, there was no significant main effect for the biological movement (F (1, 276) = 0.35, *p* = 0.55).

## 5. Discussion

Results of the experiments revealed that the effective factors for movement interference and the synchronization index are different. In the following subsections, the possible reasons for the differences are discussed based on each factor.

### 5.1. Number of Agents

As it is known, the ability of a group of people to accomplish a task efficiently is known to be enhanced when they have a similar efficiency level [41,42]. Additionally, members of groups with different beliefs are more likely to change their beliefs when the rest of the group gradually changes their own [43]. Moreover, emotions are easily spread within a group when there are more people [44]. Thus, by increasing the number of agents, the participants might perceive themselves as part of the group of agents, effectively affecting their hand movement.

However, there was no evidence that the two-agent set-up increased interference compared to the one-agent set-up, nor that the three-agent set-up increased interference compared to the two-agent set-up. The reason for not having a significant difference might be because in a two-agent set-up the number of the agents were not conceiving themselves as a group. It is possible that in the two-agent set-up the two agents were perceived as two individuals instead of a single group due to the small number of members. Therefore, it is suggested that for recognizing a group, a minimum of three agents is required for affecting the movement interference.

Nevertheless, the effect of the number of agents was not observed for the synchronization. In the interference experiment, the movement of the subject and agent(s) was orthogonal, while for the synchronization experiment it was parallel. When subjects are looking at orthogonal movements, they may focus less on the rhythm (since it appears to be difficult to follow two or more orthogonal movements by eye) and place a larger emphasis on the properties of the projected agent (such as the number of projected agents). In contrast, in the case of the parallel movement, the subjects might focus on the rhythm of the movements (since focusing on and following the parallel movements by eye seems not to be so difficult) but not on the features of the projected agents. As a result, for the interference experiment, the group of agents might be recognized easily by the participants, while for the synchronization experiment it could be difficult. This could be the main reason that the factor of the number of agents worked for the interference but did not appear in the synchronization experiment.

### 5.2. Type of Avatar

The effect of the type of avatar may be due to the perception of the subject regarding the agent as a peer when it has similar physical aspects performing the same action. As it is known, people feel more motivated to use avatars when the avatars share similar characteristics with them [45], and humans are more willing to cooperate by synchronizing their starting/ending actions in cooperative spaces when they share similar characteristics [46]. Therefore, in the synchronization experiment, a more similar agent, i.e., the human agent, led to more synchronized movement by the participant.

### 5.3. Biological Movement

The effect of the biological movement was not observed in any of the experiments. Even though Kilner [8] proposed that biological movements might have a significant effect on the interference of the moving hand, in our case it seems that we combined too many conditions. Therefore, the difference in movement seems less obvious to the participant. As a consequence, neither the movement interference nor the antiphase synchronization was affected by the agent’s biological movement.

### 5.4. Synchronization Index

Previous studies verified the effect of biological movement between robots and humans [8,9,10], the robotic shape influence [40], and humanoid robots’ influence [41]. However, they did not study how the agents influence the synchronization of the action. This synchronization influence needs more research to clarify the details of the nature of its influence. In this study, our proposed index is a phase-based index that was designed for this experiment. To study the synchronization concept more deeply, more studies with different measurements might be beneficial for understanding the dimensions of the influence of the agents on this field.

## 6. Limitations

Although in this experiment we explored the influence of the number of agents and the type of movements, we did not test any social behavior of the robots, such as speech, gaze, behavior, or relational interaction among the agents. Subjects might have been affected more significantly if such social behaviors had been included. Moreover, fatigue may influence the results of movement interference. In addition, even though we experimented using counterbalance, the subjects participated in all the set-ups, therefore possibly being influenced by previous conditions.

## 7. Conclusions

This study proposed that a group of projected agents moving their hands orthogonally to the subject would be able to increase movement interference. The projected agents were: one, two, or three agents; humans or robots; and biological agent movements or linear agent movements. To explore the effect of the proposed factors, the following comparisons were made: (1) the effect of exposing subjects to a different number of agents, different types of avatars, and different types of movements in incongruent vertical or horizontal subject movements; (2) the effect of exposing subjects to a different number of agents, different types of avatars, and different types of movements in congruent vertical or horizontal subject movements. In these experiments, we gathered 24 participants that were involved in both experiments that lasted around 25 min each. The interference of the agent on the hand movement of the participants was studied, which was around 13 mm. In addition, the synchronization of the agent on the hand movement of the participant was studied, which was around 12 deg. Increased movement interference was found between the three-robot set-up and the one-robot set-up. Enhanced synchronization was also found between the human set-up and robot set-up. The results showed that the number of robots enhanced the interference movement, which reveals knowledge regarding the influence that number of agents has on the interference of the movement of people. Moreover, we proposed an additional paradigm that was not previously considered, which was the influence on synchronization, in which we found that the avatar type influenced the ability to maintain synchronization. The results reveal knowledge regarding the influence that the avatar might have on influencing people in performing complex tasks while maintaining synchronization. However, this experiment did not verify the effect of different social behaviors of agents or different types of non-humanoid avatars, which will be analyzed in forthcoming experiments. The effect of group dynamics on people regarding the interference of synchronization and movement is also worth evaluating. The combinations of different types of non-humanoid avatars would help to understand these group dynamics in depth.

## Figures and Tables

**Figure 1 sensors-22-05465-f001:**
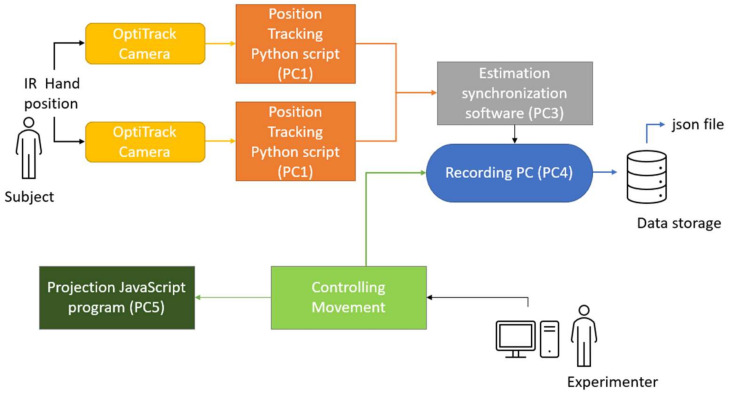
System block diagram implemented for recording hand position movement.

**Figure 2 sensors-22-05465-f002:**
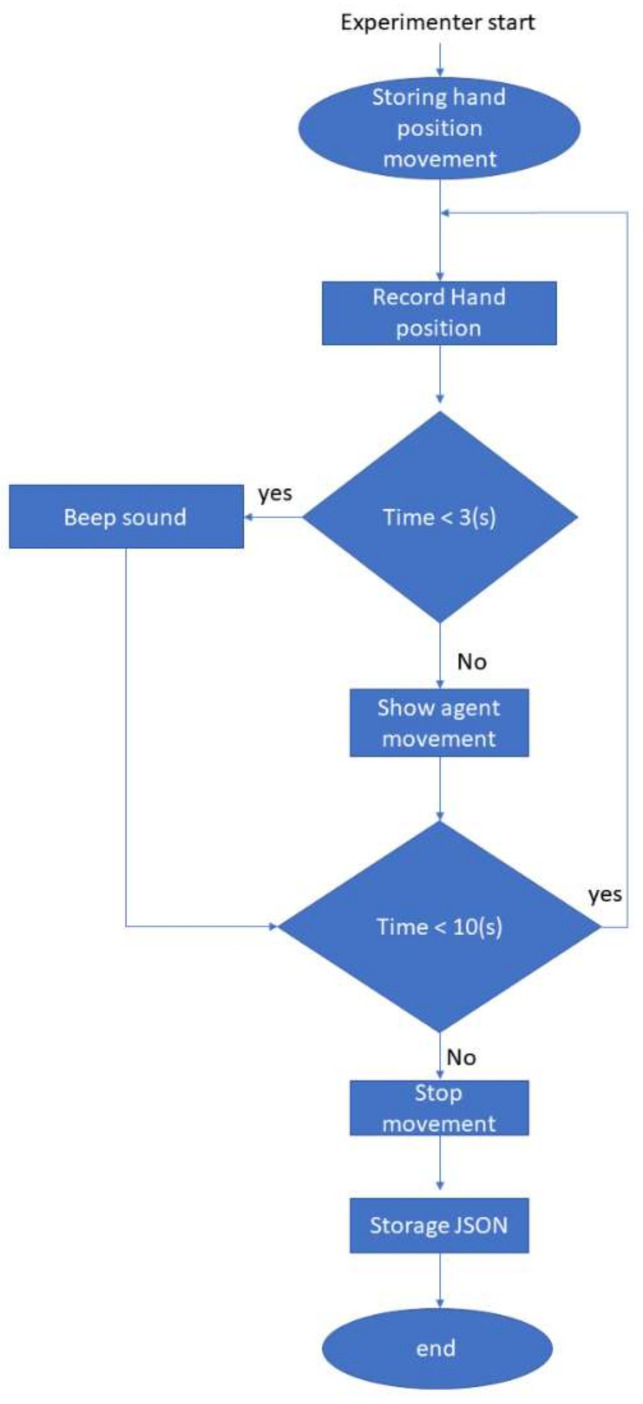
Algorithm system flow of the system for showing agent movement.

**Figure 3 sensors-22-05465-f003:**
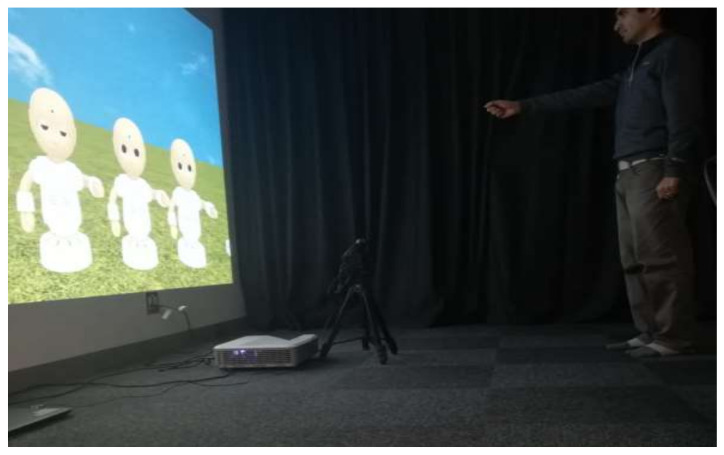
A person standing in front of the projection.

**Figure 4 sensors-22-05465-f004:**
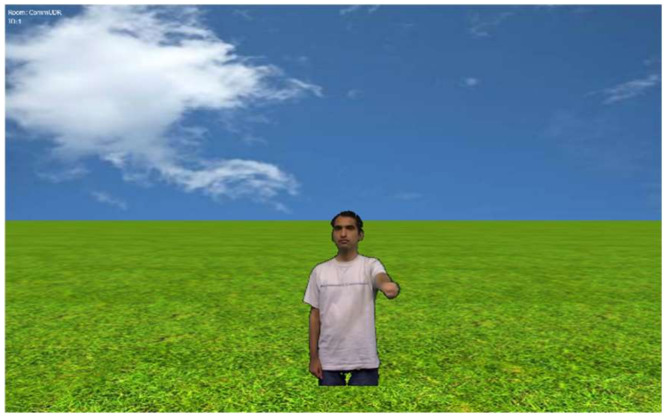
Human avatar projected in the experiment.

**Figure 5 sensors-22-05465-f005:**
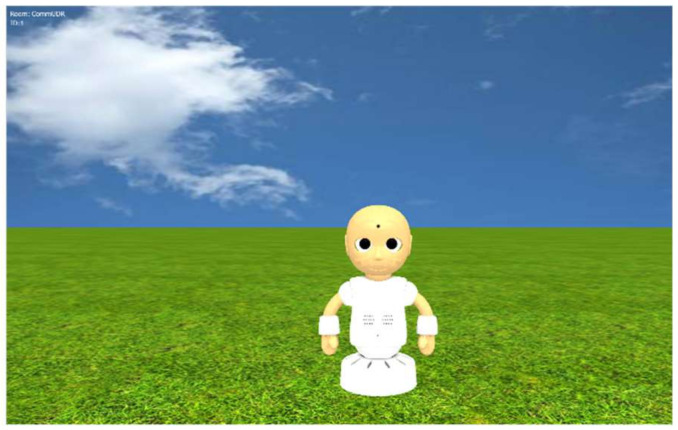
Robot avatar projected in the experiment.

**Figure 6 sensors-22-05465-f006:**
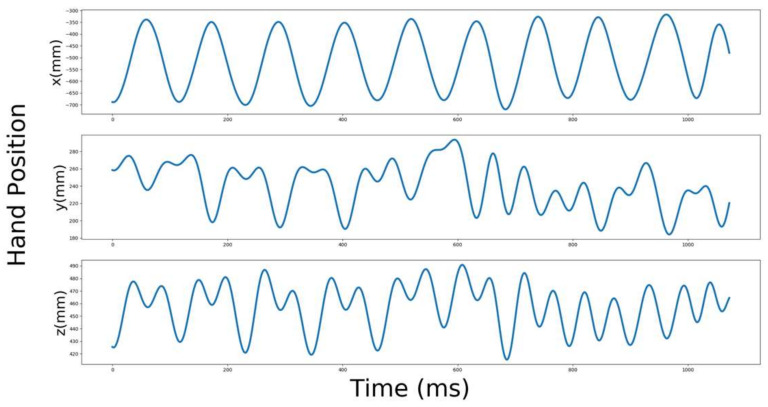
Recorded movement of the subject hand position. The top graph shows the position of the hand on the x-axis in time (ms), the middle graph shows the position of the hand on the y-axis in time (ms) and the bottom graph shows the position of the hand on the z-axis in time (ms).

**Figure 7 sensors-22-05465-f007:**
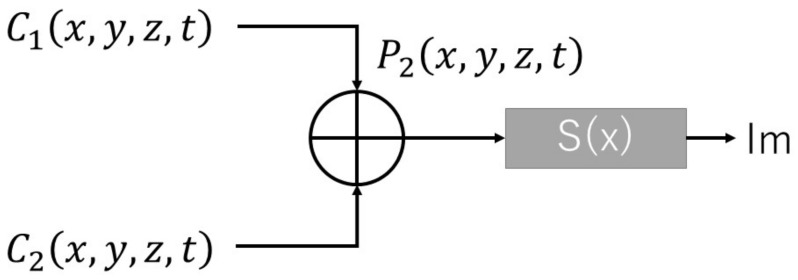
Process of combining data, calculating the standard deviation S(x) as Formula (2) shows, where x is the orthogonal axis and Im is the interference movement index.

**Figure 8 sensors-22-05465-f008:**
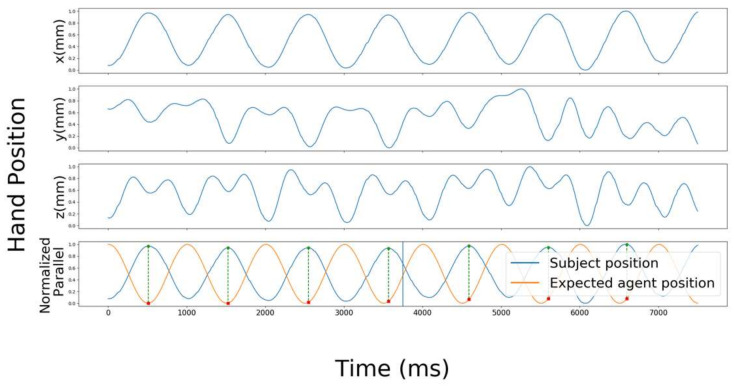
Recorded movement of a subject hand position and expected movement of the agent. The top three graphs show the standardized hand positions on the x, y, and z-axis, respectively. In the fourth graph, the standardized hand position on the x-axis (the same as the one in the first graph) is shown as a blue trajectory while the expected agent hand position is shown as an orange one. Circles indicate the peaks of the trajectory of the standardized hand position while the squares indicate their projected points on the trajectory of the expected agent position. The fourth graph shows the subject’s hand trajectory in which the first pick projection phase value is minus ninety degrees difference.

**Figure 9 sensors-22-05465-f009:**
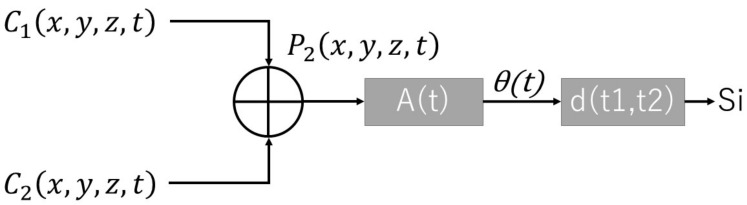
Process of combining data, calculation of the agent peak hand position times into the subject agent phase (Formula (3)) and calculating the difference between the average of phases (Formula (4)).

**Figure 10 sensors-22-05465-f010:**
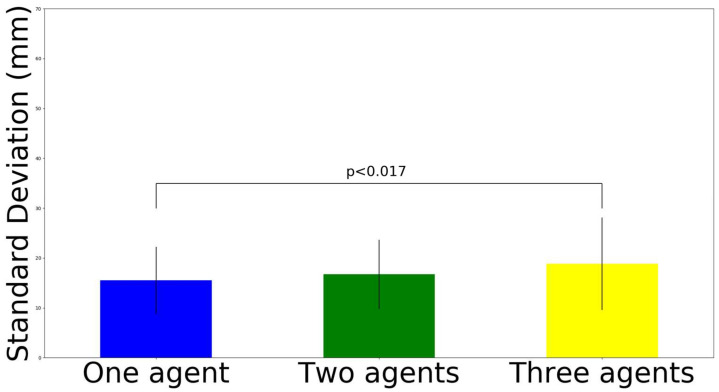
Standard deviation results on orthogonal horizontal movement interference.

**Figure 11 sensors-22-05465-f011:**
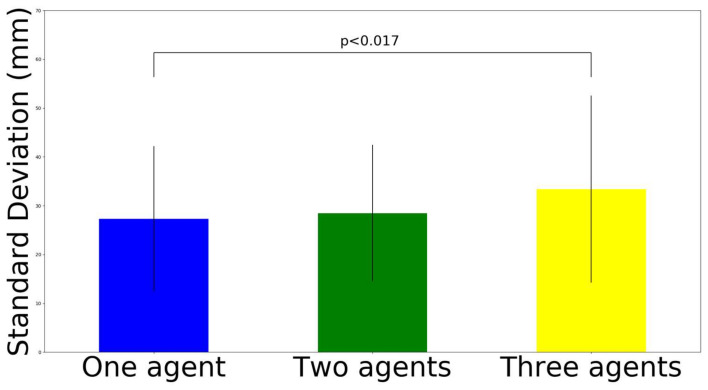
Standard deviation results on orthogonal vertical movement interference.

**Figure 12 sensors-22-05465-f012:**
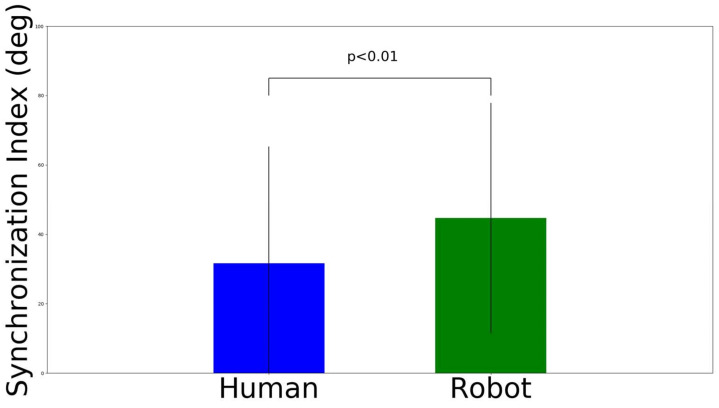
Synchronization index results on orthogonal vertical movement interference.

**Figure 13 sensors-22-05465-f013:**
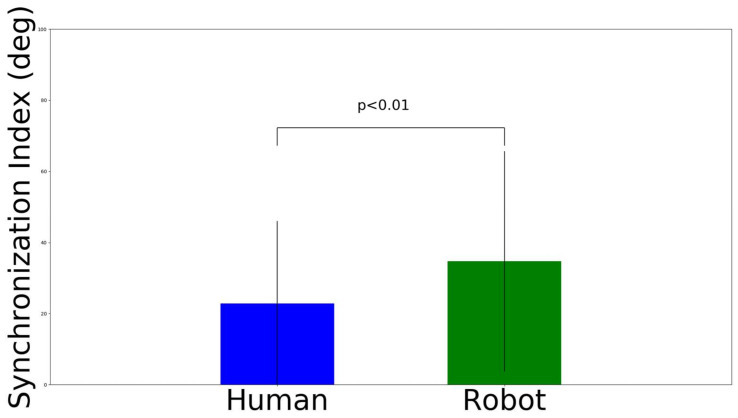
Synchronization index results on orthogonal horizontal movement interference.

**Table 1 sensors-22-05465-t001:** Three-way ANOVA for horizontal interference movement.

	DoF	Sum_sq	Mean_sq	F	PR (>F)
C (Number of agents)	2	2003.79	1001.89	3.72	0.03
C (Type of avatar)	1	12.56	12.56	0.05	0.83
C (Type of movement)	1	33.69	33.69	0.13	0.72
C (Number of agents):C (Type of avatar)	2	15.27	7.64	0.03	0.97
C (Number of agents):C (Type of movement)	2	7.14	3.57	0.01	0.99
C (Type of avatar):C (Type of movement)	1	3.56	3.56	0.01	0.91
C (Number of agents):C (Type of avatar):C (Type of movement)	2	61.87	30.93	0.11	0.89
Residual	276	74,246.1	269.01		

**Table 2 sensors-22-05465-t002:** Three-way ANOVA for vertical interference movement.

	DoF	Sum_sq	Mean_sq	F	PR (>F)
C (Number of Agents)	2	546.08	273.04	4.49	0.01
C (Type of avatar)	1	0.01	0.01	0	0.99
C (Type of movement)	1	3.75	3.75	0.06	0.8
C (Number of Agents):C (Type of avatar)	2	153.58	76.79	1.26	0.28
C (Number of Agents):C (Type of movement)	2	15.36	7.68	0.13	0.88
C (Type of avatar):C (Type of movement)	1	74.33	74.33	1.22	0.27
C (Number of Agents):C (Type of avatar):C (Type of movement)	2	35.41	17.7	0.29	0.75
Residual	276	16,790.12	60.83		

**Table 3 sensors-22-05465-t003:** Three-way ANOVA for vertical movement on time synchronization.

	DoF	Sum_sq	Mean_sq	F	PR (>F)
C (Number of agents)	2	1274.51	637.26	0.84	0.43
C (Type of avatar)	1	10,190.42	10,190.42	13.39	0
C (Type of movement)	1	803.9	803.9	1.06	0.3
C (Number of agents):C (Type of avatar)	2	383.85	191.93	0.25	0.78
C (Number of agents):C (Type of movement)	2	641.52	320.76	0.42	0.66
C (Type of avatar):C (Type of movement)	1	170.62	170.62	0.22	0.64
C (Number of agents):C (Type of avatar):C (Type of movement)	2	812.26	406.13	0.53	0.59
Residual	276	210,073	761.13		

**Table 4 sensors-22-05465-t004:** Three-way ANOVA for horizontal interference movement.

	DoF	Sum_sq	Mean_sq	F	PR (>F)
C (Number of agents)	2	274.95	137.48	0.12	0.89
C (Type of avatar)	1	12,139.16	12,139.16	10.63	0
C (Type of movement)	1	401.44	401.44	0.35	0.55
C (Number of agents):C (Type of avatar)	2	771.85	385.92	0.34	0.71
C (Number of agents):C (Type of movement)	2	25.15	12.58	0.01	0.99
C (Type of avatar):C (Type of movement)	1	2759.06	2759.06	2.42	0.12
C (Number of agents):C (Type of avatar):C (Type of movement)	2	469.99	234.99	0.21	0.81
Residual	276	315,197.4	1142.02		

## Data Availability

The data presented in this study are available on request from the corresponding author. The data are not publicly available due to the privacy of the subjects involved.

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
