# Peer review of "Multiple Groups of Agents for Increased Movement Interference and Synchronization"

_sensors, 2022, doi:10.3390/s22145465_

Round 1

Reviewer 1 Report

In this paper the authors have examined the influence of groups of agents and the type of avatar on movement interference. Also they studied the synchronization of the subject with the agent. For that, they conducted experiments with human subjects to examine the influence of one, two, or three numbers of agents, as well as human or robot avatars, and finally, the agent moving biological or linear. These results seem to suggest that the number of agents is more influential on movement interference than other avatar characteristics.

The paper describes textually the principle of the methods used in taking over the data, the equipment used and the data obtained. An extensive bibliography was also consulted.

However, apart from the very general principles of data measurement and structuring, no technical details are provided. In general, it is not clearly specified how the measured data were processed for interpretation (only textually, in short, without relations: “We divided… We calculated…”), nor how the measurements were performed from a technical point of view. The equipment was specified, but not their measurement techniques, their measurement errors, the type of sensors). The technical part was substantially reduced in the paper, being detailed the interpretation from the point of view of the connected general purpose, psychological and behavioral. The paper should be more technical-oriented, providing more technical details.

Many explanations are given in general in the text. The used techniques, would be good to be supported by relations and respectively a more elaborate technical support.

Section 4 - Results - contains many abbreviations and unexplained notations. Also,

Line 144 – “SD = 3.69” what does the abbreviation mean?

Are the cases analyzed in the paper defining for drawing general conclusions in the field?

Author Response

Dear Reviewer,

Thank you for your comments. We answered the issues you mentioned point by point in the following. Your comments are in the boxes and the answers/reactions comes after them.

In this paper the authors have examined the influence of groups of agents and the type of avatar on movement interference. Also, they studied the synchronization of the subject with the agent. For that, they conducted experiments with human subjects to examine the influence of one, two, or three numbers of agents, as well as human or robot avatars, and finally, the agent moving biological or linear. These results seem to suggest that the number of agents is more influential on movement interference than other avatar characteristics.

The paper describes textually the principle of the methods used in taking over the data, the equipment used and the data obtained. An extensive bibliography was also consulted.

However, apart from the very general principles of data measurement and structuring, no technical details are provided. In general, it is not clearly specified how the measured data were processed for interpretation (only textually, in short, without relations: “We divided… We calculated…”), nor how the measurements were performed from a technical point of view. The equipment was specified, but not their measurement techniques, their measurement errors, the type of sensors).

The technical part was substantially reduced in the paper, being detailed the interpretation from the point of view of the connected general purpose, psychological and behavioral. The paper should be more technical-oriented, providing more technical details.
Many explanations are given in general in the text. The used techniques, would be good to be supported by relations and respectively a more elaborate technical support.

We appreciate your comments about the work, and we apologize for lacking the technical details as we wanted not to make the current manuscript too much complex. In the following, we reacted to each of your concern point by point:

  1. a) For the part your mention as {it is not clearly specified how the measured data were processed for interpretation (only textually, in short, without relations: “We divided… We calculated…”)}:

 In order to clearly explain the process for the measurement, complementing the figure 2 which explain the system, we added the following figure to the part explaining the standard deviation measurement technique and synchronization index (line 259 to 268 and 364 to 370 in the new manuscript) in order to clarify the measurement technique, the errors and the processing data used.

Information added (Standard deviation)

As the figure 7 shows, we combined the signals obtained from the cameras and calculate the standard deviation of the orthogonal axis.

Figure 7. Process of combining data, calculating the standard deviation S(x) as formula 2 shows, where x is the orthogonal axis and Im is the interference movement index.

As the figure 9 shows, we combined the signals obtained from the cameras and calculate the synchronization index of the parallel axis.

Figure 9: Process of the combining data, calculation the agent peaks hand positions time into the subject agent phase (Formula 3) and calculating the difference between the average of phases (formula 4).

We also add information of our synchronization index. In order to add more technical explanation about it, we added the following sentences and formulas at line 299 to 309 of the new manuscript:

Previous Version

Current Version

To calculate the synchronization between the subject and the agent, we projected the peak values of the trajectory subject’s hand position into the expected trajectory agent’s hand position so that the first peak projection was in the minus ninety-degree phase of the trajectory agent’s hand position. We calculate the inverse sinusoidal function of every projected point to obtain the phase’s value. We divided the signals into two sections by separating the phases projection of the first half time and the latter half time. We calculated the difference between the average of the phases in the first half and the latter half. Figure 7 is shown an example of a standardized subject’s hand trajectory and the projection of the peaks on the y plane or x plane for vertical or horizontal movement respectively. The peaks were plotted with circles, and the projections with plotted with squares.

To calculate the synchronization between the subject and the agent, we projected the peak values of the trajectory subject’s hand position into the expected trajectory agent’s hand position so that the first peak projection was in the minus ninety-degree phase of the trajectory agent’s hand position. In order to obtain the phase of subject in agent hand position peaks, the inverse sinusoidal function of the subject points that time correspond the agent peak was calculated as the formula 3 shows. Figure 8 shows an example of standardized subject’s hand trajectory and the projection of the peaks on the y plane or x plane for the vertical or horizontal movement respectively.

Where:

: Phase of Subject in agent hand position peaks

We also changed and combined the following sentences in order to clarify the index calculation based on the different between the average phases to be explained with more detail on the formula 4, in line 319 to 320 of the new manuscript:

Previous Version

Current Version

We divided the signals into two sections by separating the phases projection of the first half time and the latter half time. We calculated the difference between the average of the phases in the first half and the latter half.

The signals were divided into two sections by separating the phases projection of the first half time and the latter half time as the formula 4 shows.

We calculated the difference between the average of the phases in the first half and the latter half

  1. b) For the part your mention as {nor how the measurements were performed from a technical point of view. The equipment was specified, but not their measurement techniques, their measurement errors, the type of sensors)}:

In order to precisely explain the measurement techniques, the measurement errors and type of sensors we double checked the manuscript. We added the following explanations in the lines 152 to 157 in the new manuscript as follows:

Previous Version

Current Version

We developed a system for tracking hand movement with a passive retroreflective marker. We used two Optitrack V120 trio desktop trackers for recording tracking movements. The system was able to record the passive infrared markers with a frequency of 120Hz

We developed a system for tracking hand movement. We used two Optitrack V120 trio desktop trackers for recording the position of passive infrared markers with a frequency of 120Hz. The Optitrack cameras were placed side by side with the distance of 3 meters from each other and put on a cylinder bar with the height of 1.8m. The trio camera had 3 camera sensors, with a shutter speed of 1ms, a focal lens of 3.5mm, and a low pass filter of 800nm IR. The measurement error of the hand position of the participants was around 0.8% based on the frequency of the cameras.

Section 4 - Results - contains many abbreviations and unexplained notations.
Also,Line 144 – “SD = 3.69” what does the abbreviation mean?

We apologize for the unexplained notions. We checked the section 4 -Results- and double-checked the abbreviations we utilized in the section. In order to have clear explanation for all of them, we added the explanation about the frequently used abbreviations and notations in the beginning of the section 4 – Results- as follows:

Current version

In the following result report, M means mean, SD means standard deviation and p is the calculated probability of the alpha of the utilized t-test.

We wanted to mention the standard deviation of the age in line 144, but it might be confusing due to mentioning mean age at the beginning of the sentence instead of its abbreviation M. In order to avoid this confusion, we changed the SD by the word “standard deviation”. We checked the line 144 and we realized that the following ones were used without explanation, therefore we edited line 146 of the new manuscript as follows:

 Materials and method

Previous version

Current version

Twenty-four subjects participated in all the conditions (13 females and 11 males, mean age=22.17, SD=3.69)

Twenty-four subjects participated in all the conditions (13 females and 11 males, age’s mean=22.17 years, age’s standard deviation=3.69)

Are the cases analyzed in the paper defining for drawing general conclusions in the field?

The study about the interference of the agents on the interacting human was studied widely by various previous works [Kilner, Oztop, etc.] as mentioned in the manuscript and discussed several conclusions in the field.  In the cases analyzed in the paper, we studied the lacking part of the previous works. To clarify the contribution of the current study in the field, we added the following sentences to the conclusion section of the manuscript as follows from line 486 to line 496 in the new manuscript as follows:

Previous version

Current version

In these experiments, increased movement interference was found between the three robots’ set-up and the one robot set-up. It was also found, that enhanced synchronization between human set-up and robot set-up. These results show the potential of the proposed group of agents for influencing human interaction. However, this experiment did not verify the effect of different social behaviors of agents or different types of non-humanoid avatars, which will be analyzed in forthcoming experiments.

The results showed that the numbers of robots enhanced the interference movement which reveals the knowledge about the influence that number of agents in the interference of the movement of people. Moreover, we propose an additional paradigm that was not previously considered which was the influence on the synchronization in which we found that the avatar type influence on keeping the synchronization. The results reveal the knowledge about the influence that the avatar might have on influencing people for performing complex tasks such keep synchronization. However, this experiment did not verify the effect of different social behaviors of agents or different types of non-humanoid avatars, which will be analyzed in forthcoming experiments. It is also worth to evaluate the group dynamics on the interference of synchronization and movement on people. The combinations of different type of non-humanoid avatars would help to understand these group dynamics on depth.

Reviewer 2 Report

Dear Authors 
From the summary, I would move the first two sentences to chapter two. In the abstract, the authors should focus on their achievements. The introductions and related work were done at a good level.

The group notation of citations only raises concerns. Materials and Methods are described well enough. Justify the numbering of equation 1 to the right.

Add a period at the end of the sentence in Chapter 3, first paragraph, above 3.1.Subjects, line 141. Add a space after the text before citing, eg in the places: ... represented [31,32], line 80.... to the person [34,35], line 87, people[44], etc. Check the entire manuscript.

Do not end the chapter or subsection with the figures 2,5,11 etc.  or equations (2), (3) alone, add text.  Figs. 4 and 5 are disproportionate to the others, consider whether they are not included in one drawing with a, b. The fonts in all pictures should be of the same type and size. Follow the instructions for preparing the manuscript. Separate caption Figure 6, Figure 7 from text of manuscript.

Now it is content of the manuscript. It is currently connected. The results were described well. Appropriate statistical analyzes were used. Please note the units and add them where appropriate. Please keep the proportions of the graphs and the fonts (chapter results) in relation to all the presented figures. Move the caption under fig. 11.

The discussion of the results does not raise any objections. Conclusions can be expanded a bit. Directions for further research and analysis can be indicated. Conclusions should include not only qualitative but also quantitative effects.

Achievements of prof. Hiroshi Ishiguro and his scientific group and the implementation of the publications as part of the grants guarantee their high quality of manuscript. The list of publications and their references are relevant, actual and correct.

Reviewer. 

Author Response

Dear Reviewer, we answered your concerns point by point in the followings, please check them. Your comments are in the boxes and the answers/reactions are written after them. Thank you.

Dear Authors 
From the summary, I would move the first two sentences to chapter two. In the abstract, the authors should focus on their achievements. The introductions and related work were done at a good level.

We appreciate your kind comments.

We checked the first two sentences of the abstract and since they are about the same phenomena of others previous work, we moved them to chapter two as you kindly mentioned.

For the abstract, we double-checked it carefully and applied the following edition to emphasize the achievements:

Previous version

Current version

Movement interference is a phenomenon that has previously been studied using humans or robots. However, these studies did not investigate the effects of the interference of groups (number of agents in groups) and the type of agents (human or robot). We have therefore examined the influence of groups of agents and the type of avatar on movement interference. In addition, we studied the synchronization of the subject with the agent. For that, we conducted experiments utilizing human subjects to examine the influence of one, two, or three numbers of agents, as well as human or robot avatars, and finally, the agent moving biological or linear. We found the main effect of the number of agents on the movement interference; namely, three agents had significantly more influence on the movement interference than one agent. These results suggest that the number of agents is more influential on movement interference than other avatar characteristics. For the synchronization, the main effect of the type of the agent was reviled, showing that the human agent kept more synchronization compared to the robotic agent.

We examined the influence of groups of agents and the type of avatar on movement interference. In addition, we studied the synchronization of the subject with the agent. For that, we conducted experiments utilizing human subjects to examine the influence of one, two, or three numbers of agents, as well as human or robot avatars, and finally, the agent moving biological or linear. We found the main effect of the number of agents on the movement interference; namely, three agents had significantly more influence on the movement interference than one agent. These results suggest that the number of agents is more influential on movement interference than other avatar characteristics. For the synchronization, the main effect of the type of the agent was reviled, showing that the human agent kept more synchronization compared to the robotic agent. In this experiment, we introduced an additional paradigm on the interference which we called synchronization, discovering that group of agents are able to influence on this behavioral level as well.

The group notation of citations only raises concerns. Materials and Methods are described well enough. Justify the numbering of equation 1 to the right. Add a period at the end of the sentence in Chapter 3, first paragraph, above 3.1.Subjects, line 141. Add a space after the text before citing, eg in the places: ... represented [31,32], line 80.... to the person [34,35], line 87, people[44], etc. Check the entire manuscript.

We appreciate your precise and detailed reviewing.

The equation 1 was visually moved to the right as you kindly mentioned.

We double-checked the entire manuscript as you kindly suggested, and edited the following parts, including the points you mentioned above:

  • We added the missing periods at the end of the sentence you mentioned as well as the missing spaces before the citations.
  • We added a space before the citations inline at the following lines: 25, 27, 35, 46, 47, 56, 58, 61, 63, 80, 87, 365, 368, 396, 397, 409 in the new manuscript.
  • We also added periods to all the sentences missing at the following lines: 186, 197, 205, 216, 314, 328, 339, 343, 357, 391.

Do not end the chapter or subsection with the figures 2,5,11 etc.  or equations (2), (3) alone, add text. 

We apologize for ending the chapters without text, we modify the manuscript where we ended with figures:

After the line 183, we move the figure 2.

We added the following text after figure 5.

The controlling of the avatar movements was adjusted to start at same time and end at same time. The human avatar was able to move with the same speed of the robotic avatar for biological and linear movement.

We added the following text after equation 2

In other words, the interference movement was the standard deviation of the incongruent axis of the position of the marker of the hand.

We carefully verify not ending the text alone. The figures of the results might be not correctly formatted on the pdf conversion, we verify to carefully not ending chapter without caption as well.

Figs. 4 and 5 are disproportionate to the others, consider whether they are not included in one drawing with a, b

We apologize for the non-proportioned figures. We changed the figures for having similar size as the others figures on different conditions.

The fonts in all pictures should be of the same type and size. Follow the instructions for preparing the manuscript. Separate caption Figure 6, Figure 7 from text of manuscript.

We apologize for not having the same font type and font size on the figures. We modified the figures in order to have same font type and same font size. For the caption of Figure 6 and 7, we separated the caption and included inside the figure.

Now it is content of the manuscript. It is currently connected. The results were described well. Appropriate statistical analyzes were used. Please note the units and add them where appropriate. Please keep the proportions of the graphs and the fonts (chapter results) in relation to all the presented figures. Move the caption under fig. 11.

We appreciated your comment about the description of the results. For the units, we checked again the whole of the section, and added the appropriate units at the following lines of the new manuscript:

Previous version

Current version

(M=33.38, SD=19.15) than the one agent set-up (M=27.3, SD=14.86), where t (190) =2.46, p=.015, Cohen’s d=0.35. In contrast, the two agent set-up (M=28.46, SD=13.98) was not significantly different from the one agent set-up (M=27.3, SD=14.86), where t (190) =0.56, p=.578, Cohen’s d=0.08; similarly, the three agent set-up (M=33.38, SD=19.15) was not significantly different from the two agent set-up (M=28.46, SD=13.98), where t (190) =2.03, p=.043, Cohen’s d=0.29.

The results show that three agent set-up has greater interference in the movement (M=33.38mm, SD=19.15mm) than the one agent set-up (M=27.3mm, SD=14.86mm), where t (190) =2.46, p=.015, Cohen’s d=0.35. In contrast, the two agent set-up (M=28.46mm, SD=13.98mm) was not significantly different from the one agent set-up (M=27.3mm, SD=14.86mm), where t (190) =0.56, p=.578, Cohen’s d=0.08; similarly, the three agent set-up (M=33.38mm, SD=19.15mm) was not significantly different from the two agent set-up (M=28.46mm, SD=13.98mm), where t (190) =2.03, p=.043, Cohen’s d=0.29.

The results show that three agent set-up has greater interference in the movement (M=18.82, SD=9.28) than the one agent set-up (M=15.49, SD=6.76), where t (190) =2.85, p=.005, Cohen’s d=0.41. In contrast, the two agent set-up (M=16.72, SD=6.93) was not significantly different from the one agent set-up (M=15.49, SD=6.76), where t (190) =0.18, p=.215, Cohen’s d=0.18; similarly, three agent set-up (M=18.82, SD=9.28) was not significant different from the two agent set-up (M=16.72, SD=6.93), where t (190) =1.78, p=.076, Cohen’s d=0.26.

The results show that three agent set-up has greater interference in the movement (M=18.82mm, SD=9.28mm) than the one agent set-up (M=15.49mm, SD=6.76mm), where t (190) =2.85, p=.005, Cohen’s d=0.41. In contrast, the two agent set-up (M=16.72mm, SD=6.93mm) was not significantly different from the one agent set-up (M=15.49mm, SD=6.76mm), where t (190) =0.18, p=.215, Cohen’s d=0.18; similarly, three agent set-up (M=18.82mm, SD=9.28mm) was not significant different from the two agent set-up (M=16.72mm, SD=6.93mm), where t (190) =1.78, p=.076, Cohen’s d=0.26.

The results show that robot has greater phase change in the hand movement (M=44.66, SD=33.24) than the human agent set-up (M=31.68, SD=33.64), where t (286) =3.29, p<0.01; Cohen’s d=0.39. In contrast, there was no signicant main effect for the agent number (F (2,276) = 0.84, p=0.43); similarly, there was no significant main effect for the biological movement (F (1,276) = 1.06, p=0.3)

The results show that robot has greater phase change in the hand movement (M=44.66 deg, SD=33.24 deg) than the human agent set-up (M=31.68 deg, SD=33.64 deg), where t (286) =3.29, p<0.01; Cohen’s d=0.39. In contrast, there was no significant main effect for the agent number (F (2,276) = 0.84, p=0.43); similarly, there was no significant main effect for the biological movement (F (1,276) = 1.06, p=0.3).

The results show that robot has greater phase change in the hand movement (M=34.72, SD=30.99) than the human agent set-up (M=22.82, SD=23.18), where t (286) =3.69, p<0.01; Cohen’s d=0.39. In contrast, there was no significant main effect for the agent number (F (2,276) = 0.12, p=0.89); similarly, there was no signicant main effect for the biological movement (F (1,276) = 0.35, p=0.55).

The results show that robot has greater phase change in the hand movement (M=34.72 deg, SD=30.99 deg) than the human agent set-up (M=22.82 deg, SD=23.18 deg), where t (286) =3.69, p<0.01; Cohen’s d=0.39. In contrast, there was no significant main effect for the agent number (F (2,276) = 0.12, p=0.89); similarly, there was no significant main effect for the biological movement (F (1,276) = 0.35, p=0.55).

For the proportions of the graphs, we corrected them as answered in the previous comments.

For fig11, we double-checked and realized its incorrect position, so we moved the caption to the bottom part as you kindly mentioned.

The discussion of the results does not raise any objections. Conclusions can be expanded a bit. Directions for further research and analysis can be indicated. Conclusions should include not only qualitative but also quantitative effects.

We appreciate your kind comment about the discussion. For the conclusion, the content may not include enough explanation based on the result including the quantitative reports therefore, we expanded the mentioned points by indicating the directions for further research and analysis, as well as the quantitative effects as follows:

Previous Version

Current Version

In these experiments, increased movement interference was found between the three robots’ set-up and the one robot set-up. It was also found, that enhanced synchronization between human set-up and robot set-up. These results show the potential of the proposed group of agents for influencing human interaction. However, this experiment did not verify the effect of different social behaviors of agents or different types of non-humanoid avatars, which will be analyzed in forthcoming experiments.

In these experiments, we gathered 24 participants that were involved on both experiments that last around 25 minutes each. The interference of the agent on the hand movement of the participants was studied which was around 13 mm. Also, the synchronization of the agent on the hand movement of the participant was studied which was around 12 deg. Increased movement interference was found between the three robots’ set-up and the one robot set-up. It was also found, that enhanced synchronization between human set-up and robot set-up. The results showed that the numbers of robots enhanced the interference movement which reveals the knowledge about the influence that number of agents in the interference of the movement of people. Moreover, we propose an additional paradigm that was not previously considered which was the influence on the synchronization in which we found that the avatar type influence on keeping the synchronization. The results reveal the knowledge about the influence that the avatar might have on influencing people for performing complex tasks such keep synchronization. However, this experiment did not verify the effect of different social behaviors of agents or different types of non-humanoid avatars, which will be analyzed in forthcoming experiments. It is also worth to evaluate the group dynamics on the interference of synchronization and movement on people. The combinations of different type of non-humanoid avatars would help to understand these group dynamics on depth.

Achievements of prof. Hiroshi Ishiguro and his scientific group and the implementation of the publications as part of the grants guarantee their high quality of manuscript. The list of publications and their references are relevant, actual and correct

We appreciate your comments for improving the current manuscript format and content.

Round 2

Reviewer 1 Report

The authors responded to the comments and included additional explanations in the paper.